# Sustainable Tourism Development and Economic Growth: Bibliometric Review and Analysis

Ana León-Gómez [1],* , Daniel Ruiz-Palomo [2] , Manuel A. Fernández-Gámez [2] and Mercedes Raquel García-Revilla [3]

1    Ph.D. Tourism Program, University of Málaga, Campus Teatinos, 29071 Málaga, Spain
2    Department of Finance and Accounting, University of Málaga, Campus El Ejido, 29071 Málaga, Spain; drp@uma.es (D.R.-P.); mangel@uma.es (M.A.F.-G.)
3    Department of Tourism and Marketing, Distance University Madrid, 28400 Madrid, Spain; mercedesraquel.garcia@udima.es
*    Correspondence: ana.leon@uma.es

**Abstract:** Over the past decade, there has been a growing interest in studying the impact of sustainable tourism development on economic growth. However, despite its recent scope, the scientific literature published so far has not evaluated the performance of the scientific activity of this relationship. Consequently, this study analyzes the 668 articles published to date in the Web of Science database on the effect that sustainable tourism development has on the overall long-term progress of the economy. To this end, we carry out an analysis of the most recognized authors, regions with the highest percentage of scientific production, most influential organizations, the co-occurrence of keywords, most prominent citations, publications, and co-authorship among the most recognized authors. The results obtained show the trend and impact of the literature published to date and the established and emerging research groups. Also, they identify key research topics in a way that provides a planning framework for further research in this field.

**Keywords:** sustainable tourism; tourism development; economic growth; bibliometrics; literature review





## 1. Introduction

Tourism is a constantly growing and economically important sector on a global and local level. It is of vital economic, social, and cultural importance and offers real prospects for sustainable and inclusive development [1]. Such is the importance of this sector that the number of tourist trips made each year before the arrival of COVID-19 came to surpass the world population [2]. International tourist arrivals exceeded 1.5 billion globally in 2019, representing an increase of 3.8% year-on-year [1]. Likewise, this figure is expected to continue to rise, with a forecast of 1.8 billion international tourists by 2030 [3], although this may be revised after the impacts of the COVID-19 pandemic [4] (Figure 1).

After six decades of steady growth, tourism has been established as an indispensable engine of economic growth. In this sense, it should be noted that in 2019 the travel and tourism sector experienced a growth of 3.5%, surpassing the growth of the global economy of 2.5% for the ninth consecutive year [5]. From another perspective, we can see that the sector directly contributes 4.4% of GDP, 6.9% of employment, and 21.5% of exports of services in Organisation for Economic Co-operation and Development (OECD)countries [6]. In this regard, Figure 2 shows the 15 countries with the highest percentage of contribution to GDP in terms of the travel and tourism sector.

However, the negative impacts of COVID-19 are not only limited to the loss of human lives but also include short and long-term social, economic, and political effects [7]. A longer and more intensive COVID-19 is likely to reduce global growth to 1.5% in 2020, which is half the projected growth rate before 2020, with implications for international

tourism [8]. The International Monetary Fund predicts that the COVID-19 outbreak will cause a global recession in 2020 that could be worse than the one triggered by the global financial crisis of 2008–2009 [7]. Likewise, the COVID-19 outbreak will have serious consequences for international tourism, with decisive effects on the economic growth and prosperity of various nations [4,9].

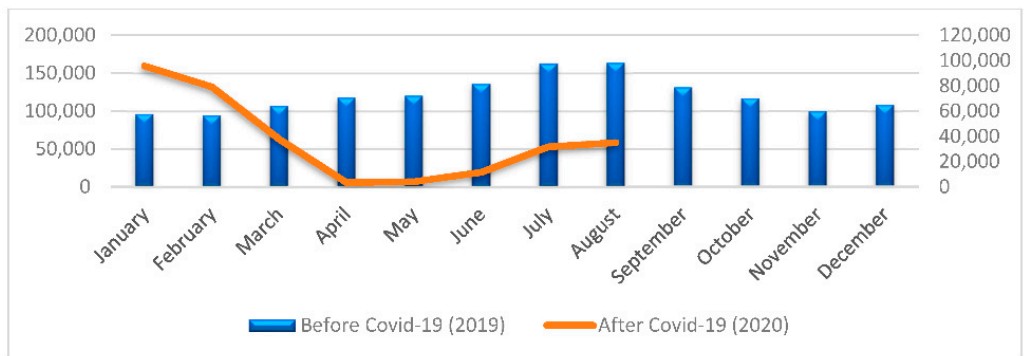

**Figure 1.** Comparison of international tourists before and after the pandemic.

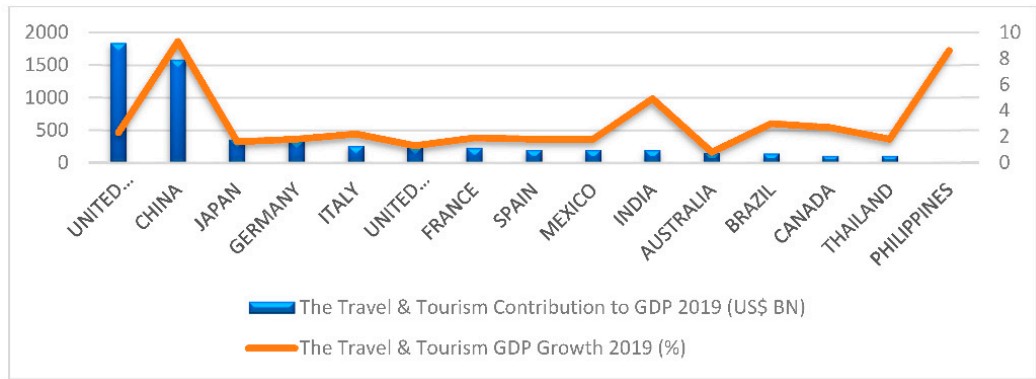

**Figure 2.** Top 15 largest countries in terms of travel and tourism GDP contribution.

The economic importance of tourism and its growth is an area that has significant value today, and therefore it is not surprising that there is a large body of literature that highlights the different impacts of tourism [2,3,7,10]. Similarly, a key aspect for tourism is sustainability, since it is considered a way to satisfy the needs of the stakeholders, taking into account economic impact as well as current and future social and environmental conditions [1,11,12]. Consequently, sustainability and the limits of its growth are a constant concern within the tourism sector [13–19].

In recent years, various reviews of the literature on sustainable tourism development and economic growth have been completed. These reviews have identified various current topics covered in the current research literature. Each study has provided information about the field examined, but further analysis of this literature using rigorous bibliometric tools may provide more information that has not been fully understood or evaluated before. In this way, bibliometric tools can be a powerful way to identify established and emerging current areas, as well as identifying research groups and researchers to show how various areas of thought may have arisen based on institutional and author characteristics. Identifying the most influential researchers within a group paves the way for determining additional emerging fields of study by pinpointing more recent topics covered by these researchers [20].

The current study, therefore, presents a comprehensive evaluation of the influence of sustainable tourism development on economic growth, starting with a group of more than 15,000 published articles and filtering this group to find specialized works on the proposed

topic. From these results, additional insights are also gained about current research interests and possible directions for future research.

## 2. Conceptual Framework

The empirical literature published so far has not specified a universally accepted definition of sustainable tourism development [13], although recent evidence on the matter suggests that one of the most widely used definitions of the concept is that provided by the World Trade Organization (WTO), which defines it as a "development that meets the needs of present tourists and host regions while protecting and enhancing opportunity for the future" [21]. However, the empirical literature continues to analyze a series of fundamental questions for the development of sustainable tourism, and many relevant questions in this area remain unresolved. Without forgetting that its genesis is associated with the maintenance of environmental quality [22,23], it is of particular importance that everything done now will not harm future generations [22]. Therefore, the assessment of long-term economic, environmental, and community health is deemed necessary [24]. Thus, recently, researchers have examined the effects of sustainability within tourism [25] since it is considered a paradigm that characterizes the future of this sector [16]. For this reason, in recent years, there has been growing interest in studying this relationship, highlighting the importance and difficulty of improving sustainability in tourism [10,26] but also identifying it as a fundamental tool for competitiveness [14,27].

Studies on economic growth demonstrate the impact of the tourist impulse on the general progress of the economy in the long term [28]. To this end, many governments have been involved in the development of tourism for economic growth, because it has a great capacity to distribute wealth, contribute to the development of emerging economies, and generate income through exports [10,28–32]. However, it can also exert long-term negative impacts on the environment [33] and damage the future economic development of tourist destinations [10]. Therefore, tourism is a transcendental tool to promote a development proposal according to sustainability criteria [10,19,34].

In recent decades, many researchers have tried to determine the effects of sustainable tourism development on economic growth [15,19,35]. In general, this approach has been used to show the growing importance of this relationship by identifying research gaps and specific areas of interest. Linking of the literature between authors, topics, and research fields has been completed through content analysis and descriptive statistics. Although we can verify that this relationship has been widely studied in the previous literature, we consider it interesting to carry out a study of the size, growth, and distribution of existing scientific documents, as well as to delve into the structure of the groups that are interested in these documents. Therefore, the present study aims to recognize the various research currents in the literature on the influence of sustainable tourism development on economic growth. In this way, the results will make it possible to identify areas of interest for current research and possible directions for future research.

## 3. Research Methodology

The main purpose of bibliometric reviews is to assess the body of existing empirical literature to determine possible research gaps and highlight the limits of knowledge [20]. The bibliometric analysis uses quantitative methods to classify data, produces representative summaries, and is recognized as a useful approach to analyzing the performance of journals, institutions, authors, and the characteristics of research topics [36]. In our study, to quantify the influence of publications, authors, and journals, we have analyzed various bibliometric indicators, including the number of publications, total citations, citations per article, the main journals, most relevant universities, and the most influential countries in the matter.

Search terms used for data collection include "Sustainable Tourism", "Tourism Development" and "Economic Growth". Three combinations of these keywords were also used, including (1) Sustainable Tourism AND Tourism Development, (2) Sustainable Tourism

AND Economic Growth, (3) Tourism Development AND Economic Growth, and (4) Sustainable Tourism, Tourism Development, AND Economic Growth.

### 3.1. Initial Search Results

We use the search "subject, title, abstract, keywords" in the Web of Science (WoS) database because it is one of the most-used databases in the academic field, compiling data on a large scale and producing statistics based on bibliometric indicators [37,38]. Initial search attempts resulted in a total of 14,993 articles. A breakdown of the search results for the four sets of keywords is shown in Table 1.

**Table 1.** The initial search results.

| Search Keywords | Search Results (No. of Papers) |
| --- | --- |
| Sustainable Tourism AND Tourism Development | 10,394 |
| Sustainable Tourism AND Economic Growth | 1132 |
| Tourism Development AND Economic Growth | 3467 |
| Sustainable Tourism AND Tourism Development AND Economic Growth | 976 |
| Total | 14,993 |

### 3.2. Refinement of Search Results

Of the 14,993 articles in Table 1, we decided to refine the search so that, in our database, only articles in English, belonging to the categories of Business Economics, Social work, Social Sciences, Environmental Sciences Ecology, Sociology, Computer Science, Engineering, Mathematics, Government Law, Social Issues, Education Educational Research, and Women Studies, appear. These adjustments reduce the total number of items to 9549. Table 2 shows the number of articles after refinement for each of the four search categories. Although we observe that Sustainable Tourism, Tourism Development, and Economic Growth are topics that are highly analyzed in the existing empirical literature, we also perceive that the combination of the three terms had hardly been studied.

**Table 2.** Search results after refinement.

| Search Keywords | Search Results (No. of Papers) |
| --- | --- |
| Sustainable Tourism AND Tourism Development | 6424 |
| Sustainable Tourism AND Economic Growth | 774 |
| Tourism Development AND Economic Growth | 2351 |
| Sustainable Tourism AND Tourism Development AND Economic Growth | 668 |
| Total | 9549 |

The concept of sustainable development has been based initially on the premise of economic growth [17]. Likewise, the sustainable development of tourism has been considered a dynamic process that constantly experiences new challenges as applied technologies and the consumption aspects of tourism change [39]. Consequently, we decided to focus our analysis on the development of sustainable tourism and economic growth, so that our research aims to develop a bibliometric study of the influence of sustainable tourism development on economic growth.

When carrying out bibliometric analysis of a research field, the first step is to evaluate the available databases, their suitability, and the consequences of the use of one or the other [40]. For this purpose, 668 articles resulting from the search for the keywords Sustainable Tourism AND Tourism Development AND Economic Growth were selected. In this way, a database with 668 references in the WoS was created, which is the basis for the empirical study (Table 2).

*3.3. Data Analysis*

To observe the evolution of the field of empirical knowledge analyzed in this research, we have adopted an inductive approach through bibliometric analysis of scientific production [41]. Likewise, classification of the literature in this study is completed with the analysis of real data through the use of a deductive approach [20]. Therefore, the purpose of this study is to combine inductive and deductive approaches through the data collection method called triangulation, incorporating different data sources and different authors (e.g., current authors, academic experts, and industry experts) [42]. Data analysis is carried out through bibliometric analysis, the results of which are presented in Section 4.2. This analysis was performed with a VOS viewer due to its ability to work efficiently with large data sets and provide a range of visualization, analysis, and innovative research [20]. It is also an effective tool for performing scientific map analysis of journal publications as it has a powerful graphical user interface and maps display capabilities [38]. For this reason, we decided to complete the bibliometric analysis with two analyses of network visualizations: map density based on co-occurrence of keywords and map density based on network data connected by co-authorship items. Keyword term co-occurrence analysis represents the number of times that two terms occur together in a set of posts [43]. For this, map density based on the co-occurrence of keywords was used. Specifically, the full count method was applied, which means that each co-occurrence link had the same weight [44]. For its part, co-authorship network analysis shows the number of publications co-authored by at least two authors [38]. The vision of density is convenient to glimpse the common structure of the authors and highlight the most significant authors in this field [45]. For this purpose, the aforementioned map density based on network data connected by co-authorship items was used. Thus, each point in the item density display has a color that indicates the consistency of items at that point. By default, the colors are blue, green, and yellow. The greater the number of elements in the vicinity of a point and the greater the weight of the contiguous elements, the closer the color of the point will be to yellow [46].

## 4. Results

*4.1. Initial Data Statistics*

Figure 3 shows the trend in the number of articles published since 2005 because this year began a growing interest in the study of the influence of sustainable tourism development on economic growth. Although this field is still in its period of growth and expansion, these results show that there is a gradual growth in publications.

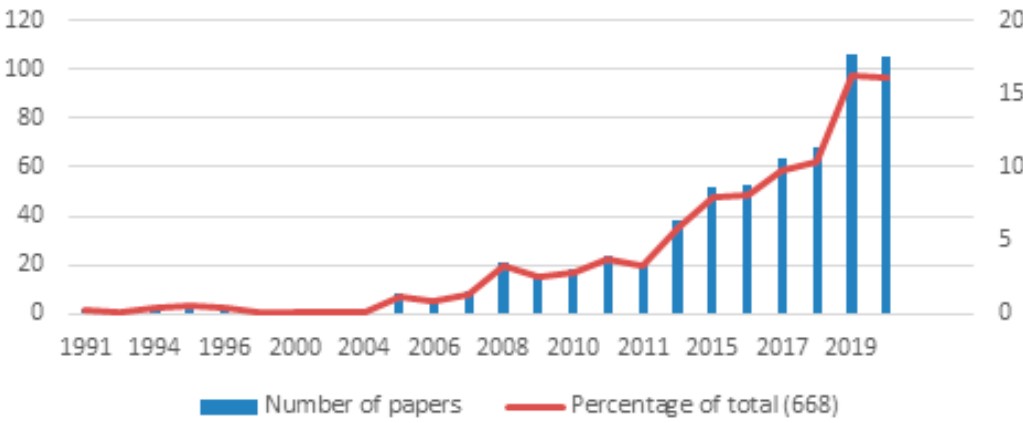

**Figure 3.** Articles are published every year.

Initial statistics show that 285 journals have contributed to the publication of 668 articles. Also, it was found that 25 journals (represented in Table 3) have published 353 articles, which represents approximately 53% of all published articles. Similarly, Appendix A

Table A1 shows the number of articles each year from the top 10 editorial journals that contribute to the area of sustainable tourism development and economic growth.

**Table 3.** The top 25 journals that contribute to the area of sustainable tourism development and economic growth.

| Titles | Records |
| --- | --- |
| Sustainability | 42 |
| Tourism Economics | 34 |
| Tourism Management | 32 |
| Current Issues in Tourism | 30 |
| Environmental Science and Pollution Research | 23 |
| Journal of Sustainable Tourism | 23 |
| Asia Pacific Journal of Tourism Research | 20 |
| Journal of Travel Research | 19 |
| Annals of Tourism Research | 17 |
| Tourism Analysis | 13 |
| Worldwide Hospitality and Tourism Themes | 13 |
| International Journal of Tourism Research | 10 |
| Anatolia International Journal of Tourism and Hospitality Research | 9 |
| Tourism Planning Development | 9 |
| Journal of Cleaner Production | 8 |
| Tourism Management Perspectives | 8 |
| Tourism Planning and Development | 8 |
| Anatolia | 7 |
| International Journal of Sustainable Development and World Ecology | 7 |
| Journal of Destination Marketing Management | 6 |
| Journal of Policy Research in Tourism Leisure and Events | 5 |
| Science of The Total Environment | 5 |
| Tourism | 5 |

*4.2. Bibliometric Analysis*

This study is inspired by the methodology used in the previous literature on bibliometrics [47–50]. This methodology has been used to perform bibliometric analyses of specific journals [18,49–51] and research areas such as tourism [10,52], Kuznets environmental curves [53], environment [54,55] and economy [56–58]. As a consequence, a systematic quantitative and qualitative evaluation of the literature of 668 WoS publications related to the study of the influence of sustainable tourism development on economic growth was carried out. In this bibliometric analysis, we determine the interconnections between articles by analyzing the frequency with which other articles cite another given article related to a specific study domain. Subsequently, the file was imported into a VOS viewer, and the most influential authors, articles, journals, research papers, institutions, and countries were extracted. Likewise, a mapping of appointments related to sustainable tourism development and economic growth was carried out. These results allowed us to explore the research streams of articles related to the influence of sustainable tourism development on economic growth. Similarly, we have carried out two network analyses to complete our study, the aforementioned map density based on co-occurrence of keywords, and the map density based on network data connected by co-authorship items.

To find the most prolific and influential authors in the field of this research, the number of articles published by each author based on the influence of sustainable tourism development on economic growth was extracted from the database. Table 4 indicates the top ten contributing authors and the number of articles in which they were authors or co-authors. The results indicate that Zaman and Croes occupy the first and second positions on the list, respectively. It should be noted that Croes and Nijkamp are co-authors of a large number of articles with each other. Likewise, Nijkamps and Romao are co-authors of the work entitled: "Spatial impacts assessment of tourism and territorial capital: A modeling study on regional development in Europe" [59].

**Table 4.** The top 10 contributing authors and the number of published articles.

| Author | Number of Published Articles |
|---|---|
| Zaman K. | 12 |
| Croes R. | 10 |
| Kumar R.R. | 8 |
| Alola A.A. | 7 |
| Sharif A. | 7 |
| Hall C.M. | 6 |
| Nijkamp P. | 6 |
| Paramati S.R. | 6 |
| Romao J. | 6 |

Although all the authors have experience in Hospitality, Leisure, Sport, and Tourism research, there are also two large subdivisions in terms of subject matter, because Hall and Nijkamp have a greater dedication in the field of Green, Sustainable Science, and Technology. In contrast, Kumar, Alola, and Romao focus their analysis on the Environmental Sciences area. Zaman, Croes, and Paramati are the authors with the greatest diversification, and their studies cover both segments. In general, the breadth of methodologies and disciplines, even among the most prolific academics, exemplifies the interdisciplinary nature of research on the influence of sustainable tourism development on economic growth.

Figure 4 shows the geographic locations of the organizations with the greatest contribution to the literature in the field of research analyzed in this study. The intensity of color in each country is proportional to the degree of participation of each organization. A higher density of contributing organizations can be found in China, with a participation percentage of 19.97%, followed by the USA (13.06%), Turkey (10.14%), England (8.14%), Spain (7.37%), Australia (6.14%), Pakistan (5.53%), Malaysia (5.38%) and Italy (4.3%). In general, the geographic dispersion of these organizations indicates that the research and practice of analyzing the contribution of sustainable tourism development to economic growth have attracted organizations and research centers from all over the world. It is also possible to make a further breakdown of these contributions for different regions. Appendix A Table A2 shows the contribution of each region to the literature on the contribution of sustainable tourism development to economic growth (note that articles with authors from different organizations may have been assigned to multiple regions).

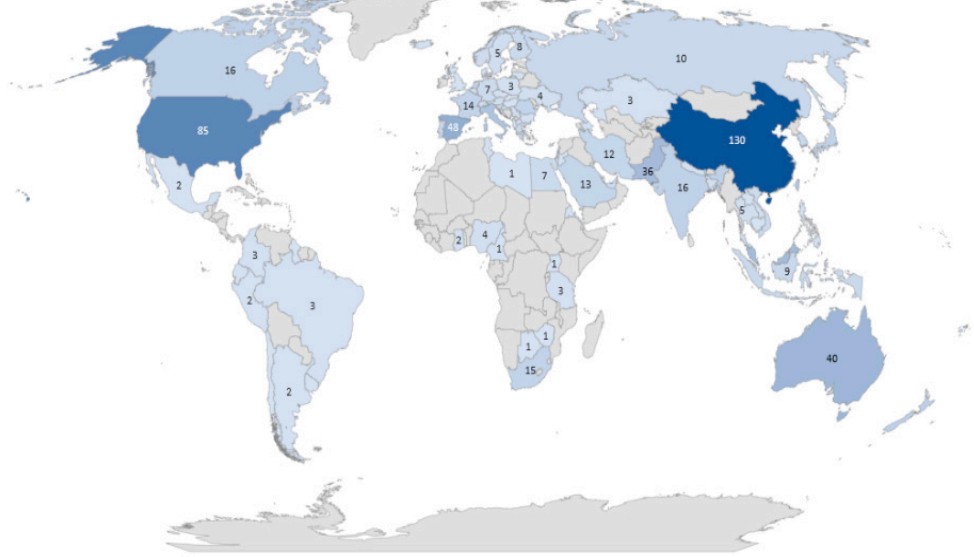

**Figure 4.** Geographic allocations of all contributing organizations.

The organizations with the best performance (according to the number of articles published), their geographic location, and the number of articles contributed are shown in Table 5.

**Table 5.** The top 20 contributing organizations.

| Organization | Location | Number of Papers |
|---|---|---|
| Eastern Mediterranean University | Cyprus | 22 |
| State University System of Florida | United States | 14 |
| University of the South Pacific | Fiji | 12 |
| Southwestern University of Finance Economics China | China | 10 |
| University Utara Malaysia | Malaysia | 10 |
| University of Central Florida | United States | 10 |
| Chinese Academy of Sciences | China | 9 |
| Hong Kong Polytechnic University | China | 9 |
| King Saud University | Saudi Arabia | 9 |
| University of Wah | Pakistan | 9 |
| University of South Carolina | United States | 9 |
| Griffith University | Australia | 8 |
| The Hong Kong Polytechnic University | China | 8 |
| Istanbul Gelisim University | Turkey | 8 |
| Bournemouth University | United Kingdom | 7 |
| SunYat Sen University | China | 7 |
| Universidade Do Algarve | Portugal | 7 |
| University of Canterbury | New Zealand | 7 |

Furthermore, in Figure 5 we can see that the continent with the highest number of contributing organizations in Asia.

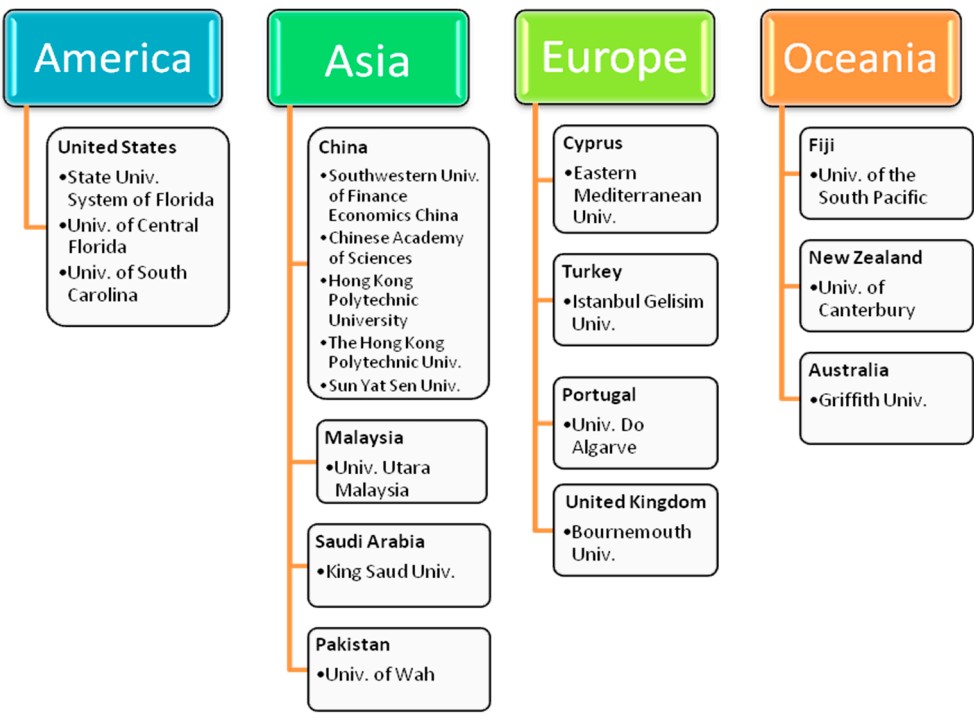

**Figure 5.** Contribution by university and region.

Table 6 shows a comparison of this list with the 10 most influential main authors previously analyzed. It is observed that the University of Wah, University of Central Florida, University of the South Pacific, Istanbul Gelisim University, Universiti Utara

Malaysia, and the University of Canterbury are represented respectively by the most prolific authors: Zaman, Croes, Kumar, Alola, Sharif, and Hall. Therefore, it may only take the work of one or two researchers for an organization to be classified as high performing.

**Table 6.** Universities of the top 10 contributing authors.

| Author | Organization |
| --- | --- |
| Zaman K. | University of Wah |
| Croes R. | University of Central Florida |
| Kumar R.R. | University of the South Pacific |
| Alola A.A. | Istanbul Gelisim University |
| Sharif A. | Universiti Utara Malaysia |
| Hall C.M. | University of Canterbury |
| Nijkamp P. | Vrije Universiteit Amsterdam |
| Paramati S.R. | Central University of Finance and Economics |
| Romao J. | Yasuda Women's University |

For its part, joint word analysis offers information on the most-studied topics and concepts [60]. This analysis links the most-used keywords in published manuscripts to describe the conceptual framework of a research field [61]. To do this, we have created a co-occurrence map of author keywords in the analysis of the influence of sustainable tourism development on economic growth (minimum threshold of 15 keyword occurrences), where 58 keywords (of the 2512 analyzed) have resulted. Figure 6 shows the evolution of the most-used terms.

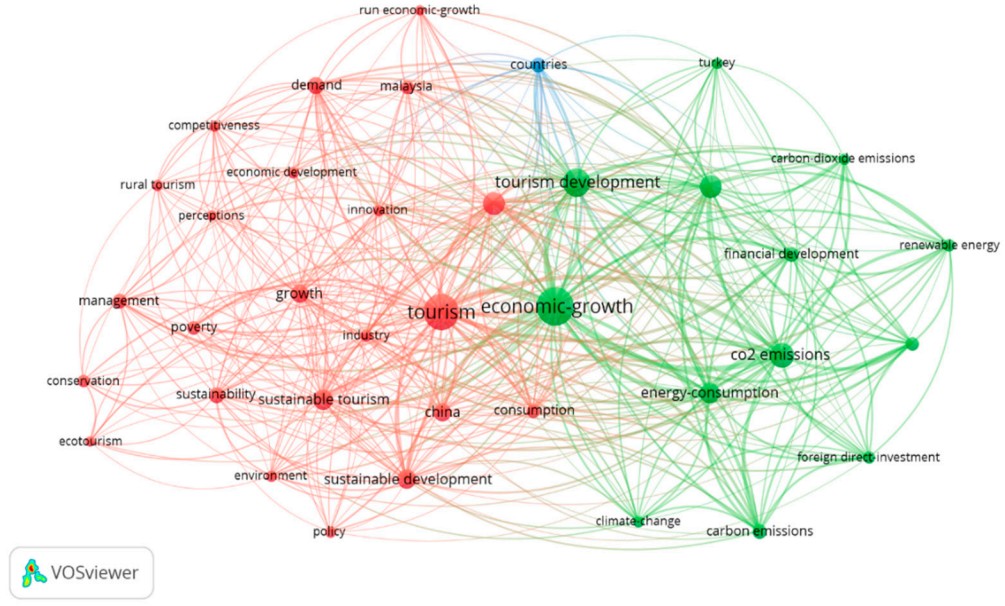

**Figure 6.** Map density based on the co-occurrence of keywords.

After analyzing a set of 881 expressions (extracted from a total of 668 articles), we made Table 7, which shows the 20 main keywords used in the articles. The keywords that appeared most frequently in the literature were: Economic Growth, Tourism, and Tourism Development. The location of these three terms on the map (Figure 6) suggests their centrality as organizing concepts in the existing empirical literature. The structure of the three central groups of the map come together to form an image of the literature that is mainly concerned with the "Sustainable Development" keyword (with an occurrence factor of 44).

**Table 7.** The most frequently used keywords in papers.

| Keyword | Occurrences |
|---|---|
| Economic Growth | 184 |
| Tourism | 161 |
| Tourism Development | 97 |
| $CO_2$ emissions | 73 |
| International Tourism | 65 |
| Impact | 61 |
| Energy Consumption | 54 |
| Sustainable Tourism | 50 |
| Sustainable Development | 44 |
| China | 42 |
| Growth | 39 |
| Demand | 36 |
| Financial Development | 29 |
| Countries | 29 |
| Carbon emissions | 29 |
| Renewable Energy | 20 |
| Consumption | 25 |
| Environmental Kuznets Curve | 24 |
| Malaysia | 23 |
| Carbon Dioxide emissions | 17 |

Finally, the 10 main articles with the highest number of citations are shown in Appendix A Table A3. The most-cited author (Hall, CM), with 612 citations, coincides with the author of the most-cited work: "Policy learning and policy failure in sustainable tourism governance: from first- and second-order to third-order change?" [62], who in turn ranks number six as a highly relevant author in the study of the influence of sustainable tourism development on economic growth. Likewise, the most recent work with the highest number of citations is: "The Effects of Tourism Economic Growth and $CO_2$ Emissions: A Comparison between Developed and Developing Economies" [63]. In Figure 7, we make a graphic comparison between the most influential authors in this field and the authors with the most citations. In this sense, we can conclude that all the authors with the most contributions are among the most-cited authors, except Sharif and Romao. Finally, Figure 8 shows us the timeline of the most relevant contributions.

Below, we also present a map based on researcher networks [46]. The elements of these networks have been connected through co-authorship links. To build the network we have used the WoS database's bibliographic files to provide them as input to the VOS viewer. Figure 9 presents the analysis of 10 related co-authors according to the number of articles published jointly. The elements included in this map are researchers, and the link between these elements is the co-authorship links between them. Likewise, each link has a strength that indicates the number of publications in which two researchers have been co-authors [46]. The total strength of the bond is described as standard weight attributes that indicate the strength of an element's bonds with others [46,64]. Four different clusters can be detected, where the largest sets consist of three elements. Zaman has the highest total bond strength, followed by Sharif, Shahbaz, Paramati, and Kumar. In turn, Zaman, Shahbaz, Paramati, and Kumar are considered influential authors in the field studied, and Zaman is the author with the greatest impact on sustainable tourism development and economic growth. In turn, these five authors are among the 13 most-cited authors in this field. Therefore, through the constructed map, it was verified that the relevance of the influence of the analyzed relationship may be due to the co-authorship between the published works. For example, Paramati and Shahbaz are especially related by their common field of research (Environmental Sciences) and have published "Does tourism degrade environmental quality? A comparative study of Eastern and Western European Union" as co-authors [65].

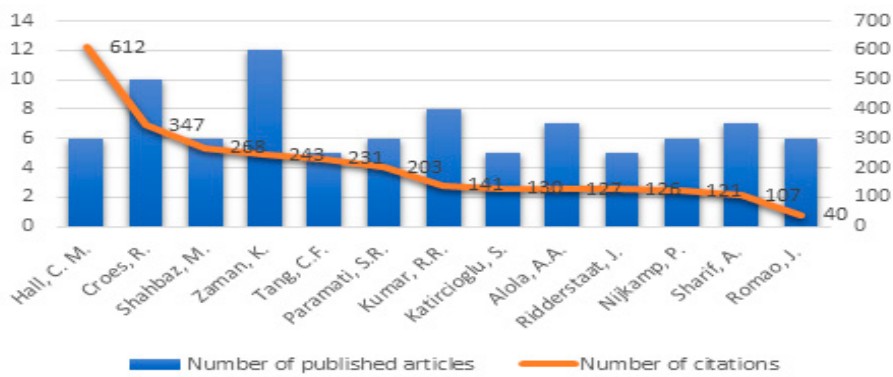

**Figure 7.** Comparison between the number of citations and number of publications.

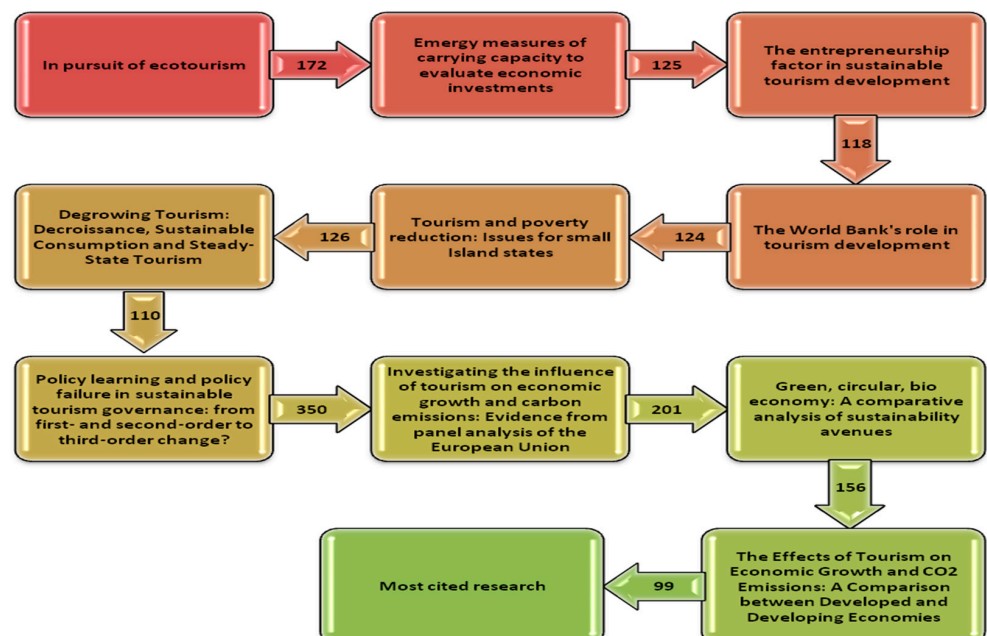

**Figure 8.** Timeline of the most influential studies.

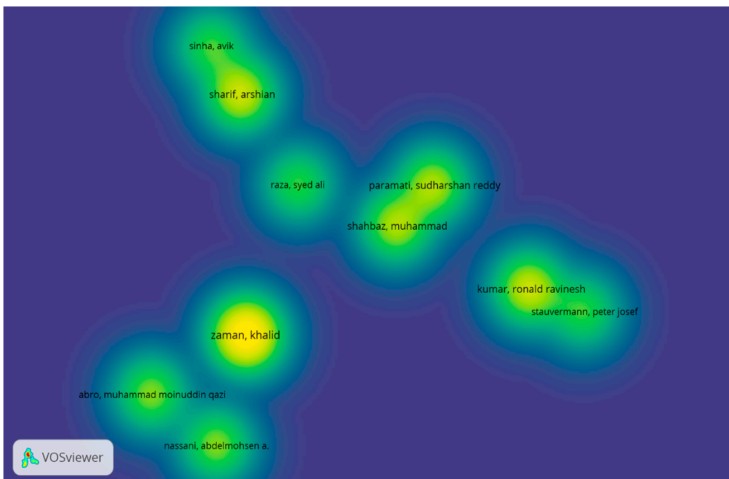

**Figure 9.** A map density based on network data connected by co-authorship items.

## 5. Discussion

This research presents a structured review of the literature that studies the influence of sustainable tourism on economic growth. Bibliometric studies on sustainable tourism and the impact of tourism on the economy are found in the previous literature [10,66,67]. However, bibliometric analysis has not been completed that relates sustainable tourism and economic growth and identifies analytically and objectively emerging works, authors, and research groups.

The results of our bibliometric analysis on sustainable tourism and economic growth indicate that there is a relative concentration of the most influential works among a certain number of researchers. Authors Zaman K., Croes R., and Kumar R.R. are the ones with the highest number of publications. Other authors also have a high number of publications, such as Sharif A., Hall C.M., Nijkamp P., and Romao J. However, as the field continues to mature, numerous authors are joining this line of research, expanding the work in a variety of areas (for example, "Green and sustainable science and technology" and "Environmental science"). As a consequence, the number of citations is progressing upwards, having increased by 28% in the last five years, which shows the current significance of the analyzed relationship. These results are in line with those obtained by previous studies on sustainable tourism [25].

Furthermore, our analysis of the geographical dispersion of publications showed that China has the highest number of works (25%), followed by the United States (20%). These results are similar to those of Yoopetch and Nimsai [19] on sustainable tourism since they indicate that this research is located in emerging regions of the world (e.g., Asia). However, our results are different from those obtained by other bibliometric studies also in the exclusive field of sustainable tourism, by indicating that the United States is the country that contributes the most literature [10,66]. This difference may be because our study also considers aspects of economic growth, which is attracting organizations, research centers, and researchers from the vast majority of countries, mainly in Asia and Europe [38].

Our study also indicates that a growing interest in the study of the influence of sustainable tourism development on economic growth began in 2005, similar to that suggested by other previous studies on economic research [38] and on sustainable tourism [10], which confirms that there is a recent and successful period of related academic literature, specifically from 2005 to date.

On the other hand, our results detect the breadth of methodologies and disciplines used, even among the most prolific academics, which exemplifies the interdisciplinary nature of research on the influence of sustainable tourism development on economic growth. Also, it is possible that only the work of one or two researchers is necessary for an organization to be classified as high performance, and that the influence of the analyzed relationship may be due to co-authorship between published works. These results coincide with those obtained by previous studies on economic research, indicating that co-authorship analysis shows that the associations between countries/regions are relatively fixed and limited [38].

Finally, the results obtained also suggest future lines of research. We have found that research on the impact of sustainable tourism on economic growth has mainly focused on the areas of hospitality, green and sustainable technology, and environmental sciences. Therefore, it would be interesting to address new areas of study that extend the results obtained. For example, the economic impact of sustainable events, sustainable tourist destinations, and sustainable maritime tourism remains to be studied. Likewise, given that current experience is concentrated in a certain number of countries, it would be necessary to expand the number of investigations into a broad set of experiences that offer a multicultural and globally relevant points of view.

## 6. Conclusions

Research on sustainable tourism development and economic growth is increasing and maturing. However, there is still a significant gap, given the small number of influential

articles. The evidence emerging from this study suggests that sustainable tourism development is indispensable for economic growth. Countries like China and the United States have developed this relationship, but for European countries, it is still an issue that needs further research. Likewise, and in the context of the current crisis derived from COVID-19, our findings suggest that measures to promote the development of sustainable tourism and thus improve economic growth, currently slowed by the pandemic, should be investigated. Additionally, we have also found that this relationship should be promoted from the point of view of environmental sciences.

On the other hand, an expansion of the results that we developed in this study seems necessary. The number of keywords could be expanded to include trade openness, pollution, energy consumption, globalization, and a wide variety of other relevant terms that could contribute to a more comprehensive review of the analyzed relationship. For example, pollution and globalization were the first precursors to the field of sustainable tourism development and economic growth. Expanding future research in this direction would identify many other contributions and potentially change core research areas even further. However, the inclusion of additional keywords will result in a broader set of articles that will later require innovative bibliometric analysis tools and approaches. Therefore, future research should consider that there are opportunities for further content analysis of specific and influential studies to more concretely identify research gaps and directions.

From an empirical perspective, the results of this study have implications for research that promotes policies to stimulate the development of sustainable tourism. This can be a solution to the scenarios of economic slowdowns, such as those caused by COVID-19 today. In this sense, there is a need to use more effective statistical models that allow for adequate decision-making with high impact on final results.

**Author Contributions:** This study has been designed and performed by all of the authors. D.R.-P. collected the data, A.L.-G. and M.A.F.-G. analyzed the data. The introduction and literature review were written by D.R.-P. and M.R.G.-R. The research methodology and initial data statistics, data analysis, and bibliometric analysis were written by M.A.F.-G. and A.L.-G. All of the authors wrote the discussion and conclusions. All authors have read and agreed to the published version of the manuscript.

**Funding:** This research was funded by the University of Malaga and the Cátedra de Economía y Finanzas Sostenibles.

**Institutional Review Board Statement:** Not applicable.

**Informed Consent Statement:** Not applicable.

**Data Availability Statement:** Web of Science (WOS).

**Acknowledgments:** The authors would like to express their sincere gratitude to the anonymous reviewers, and the editors for their truly valuable comments.

**Conflicts of Interest:** The authors declare no conflict of interest.

## Appendix A

Table A1. The top 10 publishing journals contributing to the area of sustainable tourism developments and economic growth.

| Source | Publication Year | | | | | | | | | | | | | | | | | | | | | | | | | | | | | |
|---|---|---|---|---|---|---|---|---|---|---|---|---|---|---|---|---|---|---|---|---|---|---|---|---|---|---|---|---|---|---|
| | 1991 | 1992 | 1993 | 1994 | 1995 | 1996 | 1997 | 1998 | 1999 | 2000 | 2001 | 2002 | 2003 | 2004 | 2005 | 2006 | 2007 | 2008 | 2009 | 2010 | 2011 | 2012 | 2013 | 2014 | 2015 | 2016 | 2017 | 2018 | 2019 | 2020 |
| Sustainability | | | | | | | | | | | | | | | | | | 1 | | | | | | 1 | 1 | 3 | 1 | 5 | 17 | 14 |
| Tourism Economics | | | | | | | | | | | | | | | | | | 2 | | 1 | 1 | 3 | 4 | 1 | 3 | 1 | 4 | 1 | 4 | 9 |
| Tourism Management | | | | | 1 | 1 | | | | | | 1 | | | 2 | 2 | 1 | 2 | 1 | | | 1 | 2 | 2 | 2 | 3 | 3 | 3 | | 3 |
| Current Issues in Tourism | | | | | | | | | | | | | | | | | | 1 | 1 | 3 | | | 1 | 3 | 2 | | 3 | 1 | 2 | 13 |
| Environmental Science and Pollution Research | | | | | | | | | | | | | | | | | | | | | | | | 1 | 2 | 2 | | 2 | 3 | 13 |
| Journal of Sustainable Tourism | | | | | | | | | | | | | | | | | | | | | 2 | 1 | | | 3 | 2 | 2 | 3 | 6 | 4 |
| Asia Pacific Journal of Tourism Research | | | | | | | | | | | | | | | | | | | 1 | | 1 | | 1 | 2 | 1 | 2 | 5 | 1 | 4 | 2 |
| Journal of Travel Research | | | | | | | | | | | | | | | | | | 2 | | 1 | 1 | 1 | | | 1 | 2 | 2 | 1 | 2 | 6 |
| Annals of Tourism Research | 1 | | | | 1 | 1 | | 1 | | | | | | | | | | 1 | | | | 1 | | | 1 | 3 | 2 | | 1 | 2 |
| Tourism Analysis | | | | | | | | | | | | | | | | | | | 1 | 1 | | 1 | 1 | 2 | 1 | | 2 | 1 | 2 | 1 |

**Table A2.** Contribution total of organizations based on their geographical regions.

| Geographical Region | Number of Papers | Contribution (%) |
| --- | --- | --- |
| AFRICA | | |
| South Africa | 15 | 2.304 |
| Egypt | 7 | 1.075 |
| Mauritius | 6 | 0.922 |
| Nigeria | 4 | 0.614 |
| Tanzania | 3 | 0.461 |
| Ghana | 2 | 0.307 |
| Tunisia | 2 | 0.307 |
| Botswana | 1 | 0.154 |
| Cameroon | 1 | 0.154 |
| Eritrea | 1 | 0.154 |
| Libya | 1 | 0.154 |
| Uganda | 1 | 0.154 |
| Zimbabwe | 1 | 0.154 |
| AMERICA | | |
| *North America* | | |
| United States | 85 | 13.057 |
| Canada | 16 | 2.458 |
| Mexico | 2 | 0.307 |
| Barbados | 1 | 0.154 |
| Jamaica | 1 | 0.154 |
| *South America* | | |
| Aruba | 4 | 0.614 |
| Brazil | 3 | 0.461 |
| Colombia | 3 | 0.461 |
| Argentina | 2 | 0.307 |
| Ecuador | 2 | 0.307 |
| Peru | 2 | 0.307 |
| Uruguay | 2 | 0.307 |
| Chile | 1 | 0.154 |
| Guyana | 1 | 0.154 |
| ASIA | | |
| China | 130 | 19.969 |
| Pakistan | 36 | 5.530 |
| Malaysia | 35 | 5.376 |
| Taiwan | 27 | 4.147 |
| India | 16 | 2.458 |
| Japan | 15 | 2.304 |
| Saudi Arabia | 13 | 1.997 |
| Iran | 12 | 1.843 |
| Indonesia | 9 | 1.382 |
| South Korea | 9 | 1.382 |
| United Arab Emirates | 7 | 1.075 |
| Vietnam | 7 | 1.075 |
| Singapore | 6 | 0.922 |
| Thailand | 5 | 0.768 |
| Bangladesh | 3 | 0.461 |
| Kazakhstan | 3 | 0.461 |
| Israel | 2 | 0.307 |
| Lebanon | 2 | 0.307 |
| Cambodia | 1 | 0.154 |
| Jordan | 1 | 0.154 |
| Laos | 1 | 0.154 |
| Nepal | 1 | 0.154 |
| Palestine | 1 | 0.154 |
| Philippines | 1 | 0.154 |
| EUROPE | | |
| Turkey | 66 | 10.138 |
| England | 53 | 8.141 |
| Spain | 48 | 7.373 |
| Italy | 28 | 4.301 |
| France | 14 | 2.151 |
| Romania | 14 | 2.151 |
| Netherlands | 13 | 1.997 |
| Portugal | 11 | 1.690 |
| Russia | 10 | 1.536 |
| Greece | 9 | 1.382 |
| Finland | 8 | 1.229 |

**Table A2.** *Cont.*

| Geographical Region | Number of Papers | Contribution (%) |
| --- | --- | --- |
| Germany | 7 | 1.075 |
| Scotland | 7 | 1.075 |
| Cyprus | 6 | 0.922 |
| Norway | 6 | 0.922 |
| Austria | 5 | 0.768 |
| Croatia | 5 | 0.768 |
| Serbia | 5 | 0.768 |
| Sweden | 5 | 0.768 |
| Switzerland | 5 | 0.768 |
| Slovenia | 4 | 0.614 |
| Ukraine | 4 | 0.614 |
| Denmark | 3 | 0.461 |
| Poland | 3 | 0.461 |
| Wales | 3 | 0.461 |
| Azerbaijan | 2 | 0.307 |
| Herzeg-Bosnia | 2 | 0.307 |
| Bulgaria | 2 | 0.307 |
| Czech Republic | 2 | 0.307 |
| Iceland | 2 | 0.307 |
| Lithuania | 2 | 0.307 |
| Macedonia | 2 | 0.307 |
| Albania | 1 | 0.154 |
| Belgium | 1 | 0.154 |
| Hungary | 1 | 0.154 |
| Montenegro | 1 | 0.154 |
| Serbia | 1 | 0.154 |
| Slovakia | 1 | 0.154 |
| United Kingdom | 1 | 0.154 |
| OCEANIA | | |
| Australia | 40 | 6.144 |
| New Zealand | 17 | 2.611 |
| Fiji | 12 | 1.843 |
| Marshall Island | 1 | 0.154 |

**Table A3.** Top 10 most-cited articles contributing to the area of sustainable tourism development and economic growth.

| The Title of the Publication | Authors | The Title of the Journal | Year | Total Citations |
| --- | --- | --- | --- | --- |
| Policy learning and policy failure in sustainable tourism governance: from first- and second-order to third-order change? | [62] | Journal of Sustainable Tourism | 2011 | 350 |
| Investigating the influence of tourism on economic growth and carbon emissions: Evidence from panel analysis of the European Union | [28] | Tourism Management | 2013 | 201 |
| In pursuit of ecotourism | [68] | Biodiversity and Conservation | 1996 | 172 |
| Green, circular, bio-economy: A comparative analysis of sustainability avenues | [69] | Journal of Cleaner Production | 2017 | 156 |
| Tourism and poverty reduction: Issues for small Island states | [70] | Tourism Geographies | 2008 | 126 |
| Emergy measures of carrying capacity to evaluate economic investments | [71] | Population and Environment | 2001 | 125 |
| The World Bank's role in tourism development | [72] | Annals of Tourism Research | 2007 | 124 |
| The entrepreneurship factor in sustainable tourism development | [73] | Journal of Cleaner Production | 2005 | 118 |
| Degrowing Tourism: Decroissance, Sustainable Consumption and Steady-State Tourism | [74] | Anatolia-International Journal of Tourism and Hospitality Research | 2009 | 110 |
| The Effects of Tourism on Economic Growth and $CO_2$ Emissions: A Comparison between Developed and Developing Economies | [63] | Journal of Travel Research | 2017 | 99 |

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
