# Peer review of "Sustainable Tourism Development and Economic Growth: Bibliometric Review and Analysis"

_sustainability, doi:10.3390/su13042270_

Round 1

Reviewer 1 Report

It is not possible to understand the meaning of the total figure in tables 1 and 2: since there are 3 items, A, B and C as selection criteria, the intersection of the 3 ABC produces 976 papers (table 1); but these should be part of the 10394 items of the AB papers, or of the 1132 of the AC selection, and so on. The same in table 2. These totals are thus misleading. Then, a set of additional criteria, D, is imposed, so the ABCD set includes 668 papers. Figure 3 seems to show that there is a growing trend in articles/year, but as the x-axis is reversed, the graph is counter-intuitive.

There is no real explanation about how with just three keywords the title is justified. It is more about 'Papers with sustainable tourism and economic growth keywords description'

Author Response

Before detailing the various modifications and improvements made to the previous version of this paper, we would like to sincerely thank the Reviewer for your comments and suggestions, which helped us to modify the work and make substantial improvements in the content and shape.
In the following pages, as response to proposals and comments made in the review process, we detail the changes and corrections from your comments.
Best regards.

Reviewer 2 Report

No comments outside the checklist. 

An interesting paper, although a bit simplistic. Well written and easy to read.

Author Response

(The authors gave the same response as above.)

Reviewer 3 Report

The present article employs bibliometric analysis to systematize the knowledge of sustainable tourism development to economic growth, as well as to highlight the ways of the past, present, and future research. Such papers are highly-important to the international research community and should be always welcomed. I think the present paper can be published in 'Sustainability' after certain improvements (see recommendations below).

  • In my opinion, this paper should be labeled as 'Review', not 'Article'.
  • Figure 1: please, make the caption more informative.
  • Section 2: avoid dot after the section title.
  • Sub-section 2.2: please, explain why you prefer WoS to Scopus.
  • Table 1: is it possible that some papers are counted twice, i.e., one paper belonging to two search parameters? If so, the total number of sources is smaller.
  • Sub-section 3.4 is out of place, this is more relevant to Results.
  • Table 4: what do mean 'editorial magazines'? Peer-reviewed journals? And what about the word 'Font'?
  • Section 4 can become a sub-section of section 3.
  • Section 5 to be named Results.
  • ATTENTION: Discussion is small and do not bear citations. This is inappropriate, and I ask you to work more with this part of the manuscript. First, you need to explain the registered patterns. Second, you need to indicate the main research themes that require bigger attention of the researchers. Third, you need to compare your findings to those from the other studies (many bibliographical surveys and reviews on tourism and sustainability have been published recently).
  • Conclusions: please, list your main findings (from Results and Discussion).
  • Please, check the numbering of the sections Discussion and Conclusions.
  • Please, polish the language and avoid sections/sub-sections consisting on only one paragraph.
  • Please, bring the style of references in order (check the journal's rules).

Author Response

(The authors gave the same response as above.)
